# Chitin-Derived AVR-48 Prevents Experimental Bronchopulmonary Dysplasia (BPD) and BPD-Associated Pulmonary Hypertension in Newborn Mice

**DOI:** 10.3390/ijms22168547

**Published:** 2021-08-09

**Authors:** Pragnya Das, Suchismita Acharya, Varsha M. Prahaladan, Ogan K. Kumova, Shadi Malaeb, Sumita Behera, Beamon Agarwal, Dale J. Christensen, Alison J. Carey, Vineet Bhandari

**Affiliations:** 1Department of Pediatrics, Division of Neonatology, Cooper University Hospital, Camden, NJ 08103, USA; das-pragnya@cooperhealth.edu (P.D.); prahaladan-varsha@cooperhealth.edu (V.M.P.); 2Department of Pediatrics, Division of Neonatology, Drexel University, Philadelphia, PA 19102, USA; snm56@drexel.edu (S.M.); ajc327@drexel.edu (A.J.C.); 3AyuVis Research, Inc., 1120 South Freeway, Fort Worth, TX 76104, USA; sacharya@ayuvis.com (S.A.); sbehera@ayuvis.com (S.B.); 4Department of Pharmacology & Neuroscience, University of North Texas Health Science Center, Fort Worth, TX 76107, USA; 5Department of Microbiology & Immunology, Drexel University, Philadelphia, PA 19129, USA; okumova@gmail.com; 6GenomeRxUS, Secane, PA 19018, USA; beagarwa@gmail.com; 7Dale J. Christensen Consulting LLC, Cary, NC 27519, USA; dale@djcbio.com; 8Department of Medicine, Division of Hematology, Duke University Medical Center, Durham, NC 27710, USA

**Keywords:** bronchopulmonary dysplasia, AVR-48, chitohexaose, inflammation, BPD-associated pulmonary hypertension

## Abstract

Bronchopulmonary dysplasia (BPD) is the most common complication of prematurity and a key contributor to the large health care burden associated with prematurity, longer hospital stays, higher hospital costs, and frequent re-hospitalizations of affected patients through the first year of life and increased resource utilization throughout childhood. This disease is associated with abnormal pulmonary function that may lead to BPD-associated pulmonary hypertension (PH), a major contributor to neonatal mortality and morbidity. In the absence of any definitive treatment options, this life-threatening disease is associated with high resource utilization during and after neonatal intensive care unit (NICU) stay. The goal of this study was to test the safety and efficacy of a small molecule derivative of chitin, AVR-48, as prophylactic therapy for preventing experimental BPD in a mouse model. Two doses of AVR-48 were delivered either intranasally (0.11 mg/kg), intraperitoneally (10 mg/kg), or intravenously (IV) (10 mg/kg) to newborn mouse pups on postnatal day (P)2 and P4. The outcomes were assessed by measuring total inflammatory cells in the broncho-alveolar lavage fluid (BALF), chord length, septal thickness, and radial alveolar counts of the alveoli, Fulton’s Index (for PH), cell proliferation and cell death by immunostaining, and markers of inflammation by Western blotting and ELISA. The bioavailability and safety of the drug were assessed by pharmacokinetic and toxicity studies in both neonatal mice and rat pups (P3-P5). Following AVR-48 treatment, alveolar simplification was improved, as evident from chord length, septal thickness, and radial alveolar counts; total inflammatory cells were decreased in the BALF; Fulton’s Index was decreased and lung inflammation and cell death were decreased, while angiogenesis and cell proliferation were increased. AVR-48 was found to be safe and the no-observed-adverse-effect level (NOAEL) in rat pups was determined to be 100 mg/kg when delivered via IV dosing with a 20-fold safety margin. With no reported toxicity and with a shorter half-life, AVR-48 is able to reverse the worsening cardiopulmonary phenotype of experimental BPD and BPD-PH, compared to controls, thus positioning it as a future drug candidate.

## 1. Introduction

Bronchopulmonary dysplasia (BPD) is a neonatal condition that occurs in infants born at <28 weeks of gestation and birth weights <1000 g, occurring in immature lungs (with or without respiratory distress syndrome (RDS) or “early BPD”) and progressing through a phase of “evolving BPD” and culminating in the final stage of “established BPD.” [1,2]. The strongest risk factors for BPD are prematurity and low birth weight [1]. Secondary to premature birth, the babies have immature lungs. While affected infants can improve over time due to lung growth, they will suffer from significant morbidity in childhood, extending up to adulthood, due to neurodevelopmental impairment, asthma, and emphysematous changes of the lung. While many drugs have been tried to prevent or attenuate BPD [3,4], no specific and effective treatment is available; therefore, this disease is still associated with high mortality and morbidity [5]. Despite improved neonatal care, the number of BPD cases due to this condition have not decreased [6], secondary to increased survival of infants of lower gestational ages. Although exogenous surfactant is standard-of-care treatment for RDS in premature neonates, there is no effective prevention or treatment for BPD to date [3]. Use of steroids as anti-inflammatory therapy is partially helpful in minimizing inflammation in BPD; however, in babies administered the drug (either ante- and post-natally via parenteral or inhaled routes), the incidence of BPD is either not decreased or the risk of death and poor neurodevelopmental outcome outweighs the overall benefit.

BPD is a multifactorial clinical syndrome of lung injury that affects normal alveolarization and microvascular development leading to anatomical changes that contribute to abnormal gas exchange and pulmonary mechanics [7]. This imbalance results in increased cell death and decreased cell proliferation associated with overall lung inflammation that contributes to the BPD phenotype. The alveoli become expanded with simplified alveolar epithelium and disrupted endothelium that interferes with the growth of distal airspace [8]. The progression toward BPD is an uncertain and unpredictable process, and there are no definitive medications available to date to reduce the risk of the progression of this disease in randomized clinical trials (RCTs) [9]. There have been no RCTs where inhaled budesonide has been used to treat “established BPD” [10]. In the largest RCT on inhaled budesonide [11] used during the “early and “evolving” phases of BPD, although there was a significant lowering of the incidence of “established” BPD [12], there was no difference in neurodevelopmental outcomes [13] and significantly increased mortality in the treatment group [14]. Trials for using inhaled budesonide, budesonide-surfactant combination, mother’s milk, or use of intramuscular vitamin A and prophylactic hydrocortisone have resulted in a modest reduction in the rate of BPD, but does not cure the disease [9,15]. A new preclinical meta-analysis has demonstrated the benefits of mesenchymal stromal cell therapy in animal models, while the results of early clinical trials are still pending [16].

BPD-associated pulmonary hypertension (BPD-PH) is a chronic inflammatory co-morbid condition with devastating short- and long- term consequences [17]. Infants with BPD are predisposed to abnormal growth of pulmonary vasculature with dysregulated pulmonary vascular density and increased pulmonary vascular resistance, which contributes to BPD-PH. The pathogenesis of BPD-PH is poorly understood and therefore there is less data currently about appropriate therapy. Animal studies and several clinical studies suggest that medications targeting the nitric oxide (NO) signaling pathway (NO inhalation, oral sildenafil citrate) may be effective treatment for BPD-PH, but they have not been specifically approved for this indication [18].

Due to their natural occurrence, non-toxic nature, and biodegradability, low molecular weight natural oligosaccharides such as chitohexaose and one of its derivatives AVR-25, derived from chitosan and chitin polymers, have gained considerable attention as therapeutic candidates for multiple inflammatory diseases [19,20,21]. After conducting a medicinal chemistry campaign and structure-activity relationship study by screening >50 compounds, we identified a chitin analog AVR-48 (2S,3R,4R,5S,6R)-4,5-dihydroxy-6-(hydroxymethyl)-2-(4-nitrophenoxy) tetrahydro-2H-pyran-3-yl) acetamide. AVR-48 was optimized to deliver promising therapeutic potential in mouse models of cecal ligation and puncture (CLP)-induced sepsis in young adult mice, lipopolysaccharide (LPS)-induced lung injury in adult mice, and hyperoxia-induced lung injury in adult and neonatal mice [22]. As BPD is an inflammatory disease characterized by high levels of cytokines, chemokines, and inflammatory cells in the lungs and blood serum and plasma, we took advantage of the anti-inflammatory property of AVR-48. We used AVR-48 as a pharmacological compound in a hyperoxia-induced experimental neonatal mouse model to evaluate its efficacy in preventing or attenuating the cardiopulmonary phenotype of BPD.

## 2. Results

### 2.1. Safety Profile of AVR-48

To assess the safety of AVR-48, two doses of intravenous (IV) slow bolus injections or subcutaneous (SC) injections or intranasal (IN) instillation were given to mice or rat pups (postnatal day 3–5 or P3-P5), >6 h apart. The total daily doses were up to 100 mg/kg/day IV, and up to 150 mg/kg/day SC, for 3 consecutive days. All doses were well-tolerated and there were no observed adverse clinical signs and or any change in body weight (data not shown). A slight decrease in white blood cell count, lymphocyte count (in females only) and total bilirubin levels (SC groups only) were noted in treated animals that were considered to be non-adverse since they were mild and not dose dependent in frequency or severity (data not shown). In the pups dosed with AVR-48 twice daily IV at 50 mg/kg/dose, a higher incidence of dermal/subcutaneous hemorrhage at the injection site was frequently associated with subcutaneous mixed cell infiltrate in the cheek, mandibular or cervical areas, as compared to animals dosed IV with vehicle only. Although uncertain, a higher incidence of these symptoms in IV dosed pups indicates that they may be drug-related vascular irritation. There was no change in any of the hematological or clinical chemistry parameters. Signs of discoloration, swelling, and macroscopic and microscopic signs of local irritation occurred at the site of administration in all treatment groups and were attributed to the administration vehicle (formulation of 10% DMSO, 20% Tetraglycol, and 20% PEG 400 in sterile water). There was no evidence of any AVR-48 related systemic gross observations at necropsy in the visceral organs of both mouse (Appendix A) and rat pups (not shown), and no adverse findings attributable to AVR-48. Based on the parameters monitored in this study, the maximum tolerated dose (MTD) and no-observed-adverse-effect level (NOAEL) were considered to be 100 mg/kg/day via IV and 150 mg/kg/day via SC routes of dosing.

### 2.2. Pharmacokinetic (PK) Profile of AVR-48

The PK studies of AVR-48 were developed and designed in-house by high performance liquid chromatography (HPLC) in both mouse and rat pups by IV, IP and IN dosing to check the bioavailability of the drug formulated as solution, suspension or nanoparticle encapsulation in plasma, broncho-alveolar lavage fluid (BALF) and lung tissues.

The first study was conducted in the plasma of mouse pups by IP and IN routes to determine the dose range for efficacy (Appendix A). A T_max_ of 0.0833 hour (h) was found for either 10 mg/kg or 0.22 mg/kg IP dose with rapid clearance from the blood with T_1/2_ of 0.36 h for the 10 mg/kg dose. The number of samples with AVR-48 levels above the lower limit of quantitation (LLOQ) did not allow for T_1/2_ determination in the 0.22 mg/kg dose (Appendix A). For IN dosing, a T_max_ of 1 h was found with a dose of 0.22 mg/kg and rapid clearance was observed by 2 h (Appendix A). The availability of AVR-48 in the lung tissues was similar to that of plasma both by IP and IN routes; a T_max_ of 0.0833 h was recorded for the 10 mg/kg dose (Appendix A; Appendix A). Low levels of AVR-48 did not allow for T_1/2_ determination in the 10 mg/kg dose. Extremely low levels of AVR-48 in the lung tissue did not allow for any PK parameters to be calculated for the 0.22 mg/kg dose. Taken together, these data suggest that AVR-48 is cleared from circulation rapidly in the mouse pups following IP injection [22] and that the observed efficacy in preventing BPD in the mouse pups by IP injection may be due to systemic exposure of AVR-48. Nevertheless, our data demonstrates the feasibility of delivering AVR-48 via IN as well as IP routes.

The second PK study was performed in rat pups wherein AVR-48 was administered by either SC or IV (Appendix A) at 100 and 150 mg/kg/day for 3 consecutive days and the maximum plasma concentration of the drug was recorded at 30 and 60 minutes (min) (T_max_) for the above two groups, respectively. For the IV dosed animals, the maximum concentration of AVR-48 in plasma declined in a bi-exponential fashion in which T_1/2_ was not estimable. For SC dosed animals only, AUC_(0-t)_ increased in a dose-proportional manner between 50 and 75 mg/kg/dose (Appendix A). The exposure to AVR-48 did not change substantially after 3 days of twice daily administration and there was no accumulation of the drug. Absolute bioavailability was estimated at (2010/5460) × 100 = 36% on Day 1 and (2610/4200) × 100 = 62% on Day 3 using AUC_(0-t)._

In the BALF, the maximum AVR-48 concentration on Day 1 ranged from 0.891 to 1.07 μg/mL (C_max_) and appeared between 2- and 15-min post dosing. The maximum BALF AVR-48 concentration on Day 3 ranged from 0.780 to 1.67 μg/mL (C_max_) and occurred between 15- and 60-min post doses (T_max_) for both routes of administration (Appendix A). A sustained level of AVR-48 was observed only where T_1/2_ was not estimable. For the IV dosed animals only, maximum concentration was followed by decline on PK Day 1 but was followed by sustained level of AVR-48 on PK Day 3. T_1/2_ was also not estimable. For SC-dosed animals only, AUC_(0-t)_ increased in a dose-proportional manner between 100 and 150 mg/kg/day, except on Day 3 where AUC_(0-t)_ decreased in a less than dose-proportional manner. In summary, exposure to AVR-48 increased for 100 mg/kg/day for both routes after 3 days of twice daily administration but not for the 150 mg/kg/day groups. The accumulation ratio using AUC_(0-t)_ for the IV dosed animals was 2.04, and for the SC-dosed animals, they were 1.90 and 0.880 for the 100 and 150 mg/kg/day doses, respectively.

### 2.3. Drug Release and Dose Response Study

The size of the poly D, L-lactic-co-glycolic acid (PLGA) encapsulated nanoparticle form of AVR-48 was determined using dynamic light scattering (DLS) and was found to be 369 ± 45 nm and a zeta potential to be −19.36 mV. There was ~60% drug release from the PLGA encapsulated AVR-48 nanosuspension in PBS_7.4_ at 37 °C during the first 12 h followed by 70–100% release over 15 days (Appendix A). We conducted a maximum dose response study using several concentrations of AVR-48 nanosuspension via IN route of dosing and determined that 0.11 mg/kg is a safe and efficacious dose. In our earlier report [22], we had demonstrated the efficacy dose of AVR-48 to be 10 mg/kg/dose via IV dosing, in an adult respiratory distress syndrome and acute lung injury (ARDS/ALI) mouse model. In the present study, the C_max_ values in the plasma of the mouse pups informed us that a single IV or IP injection of 10 mg/kg dose of AVR-48 provided C_max_ of 5.78 ± 0.91 µM and should therefore be sufficient to produce the desired anti-inflammatory therapeutic response. Hence, 10 mg/kg was selected as the optimum dose for AVR-48 to be tested in the BPD mouse model studies to subsequently conduct a dose response study to determine the minimum efficacy dose.

Simultaneously, we also tested the AVR-48 nanosuspension formulation IN (0.11 mg/kg) as well as in solution form, IP and IV (10 mg/kg) (through the facial vein) and confirmed that all routes of administration gave similar outcomes. To prove that AVR-48 reached the lungs when delivered IN, our test compound was conjugated with fluorescein isothiocyanate (FITC) and then evaluated histologically to confirm that it reached the lungs, as was evident from green fluorescent staining on lung sections (Appendix A). From the PK study, as described above, the bioavailability of the drug in the plasma and lungs were similar when delivered as solution formulation via either IN or IP routes. As the goal of our study was to develop AVR-48 as a commercially viable and applicable therapeutic candidate for BPD, we selected the IP route as the preferred mode of drug delivery in neonatal murine pups, with the rationale that it is easier to deliver the drug systemically to preterm babies via IV, versus the IN route. The results and outcome of all the routes of administration are presented in a cumulative manner, as the endpoint was similar for all routes of administration. The dose response study via IP dosing using 0.5, 2.0, 5.0 and 10 mg/kg doses demonstrated that 5.0 mg/kg is the minimum efficacious dose while 10 mg/kg was the optimum dose in preventing the BPD (>80%) phenotypes (Figure 1). The PLGA nanosuspension (0.11 mg/kg) when delivered IN also resulted in similar efficacy.

### 2.4. AVR-48 Restored Lung Morphology and Improved Alveolar Cellular Physiology

After assessment of the safety profile of the AVR-48, and confirmation of the ideal dose to be used, we next performed experiments to evaluate the therapeutic effect on the lungs in our experimental BPD mouse model, as previously described [19,23,24,25]. BPD is characterized by enlarged simplified alveoli with large air sacs, a thickened septum, and thin alveolar epithelium (Figure 2A), accompanied by an overall decrease in alveolar cell proliferation, decreased or dysregulated angiogenesis, and increased cell death. All these features were restored after injection of two doses of AVR-48, IP (10 mg/kg) on days P2 and P4. The alveolar sacs, as evident from the chord length (Figure 2B) and the septal thickness (Figure 2C), regained their normal shape and size in the treated group, comparable to that of RA controls. The radial alveolar counts (RAC), which are decreased in the diseased condition, were also improved after AVR-48 treatment (Figure 2D). The total inflammatory cells present in the BALF (Figure 3A) and the total protein content (Figure 3B)—both increase in BPD, was decreased after treatment with AVR-48. There was a revival in cell proliferation (as evident from Ki67 immunostaining; Figure 4A). To assess cell specificity, we focused on Type I and Type II alveolar epithelial cells (AECs), as they are most relevant to the process of alveolarization. We utilized the receptor for advanced glycation end-products (RAGE) as the preferred marker for Type I AECs [26,27] and surfactant protein (SP)-C for the Type II AECs [28,29]. Due to Ki67 and SP-C being raised in the same species, we used proliferating cell nuclear antigen (PCNA) expression—a well-established marker of cell proliferation because cells remain for a longer time in the G1/S phase when proliferating—as a preferred marker for proliferating cells. The number of cells co-localizing with surfactant protein (SP)-C and PCNA were less in the BPD group as compared to RA, RA+AVR-48, and BPD+AVR-48 groups (Figure 4B). There were few cells which were double positive for PCNA as well as RAGE in the RA and RA+AVR-48 groups (Figure 4C). Although the number of RAGE+ve cells were decreased in the BPD group as compared to RA, RA+AVR-48, and BPD+AVR-48 groups, these did not co-localize with PCNA in the BPD or BPD+AVR-48 groups (Figure 4C). In addition, there was a decrease in cell death (as shown by TUNEL staining (Figure 5A) and immunoblotting of cleaved caspase 3; Figure 5B) after treatment with AVR-48, as compared to the BPD group.

The blood vessels, which are usually disrupted in BPD, showed improvement (as was evident from vWF immunostaining; Figure 6A). Angiopoietin 2 (Ang2), which is increased in BPD [30,31], was significantly decreased after treatment with AVR-48 (Figure 6B), thus suggesting that AVR-48 treatment may be able to stabilize vascular leak and promote sprouting neo-angiogenesis.

### 2.5. AVR-48 Did Not Have Any Adverse Effect with Surfactant

Since exogenous surfactant is used as the standard of care in neonatal intensive care units (NICUs) to prevent and manage RDS in early life of preterm neonates, we wanted to test the impact (if any) of the concomitant use of AVR-48 with surfactant. Although mice are surfactant sufficient (dissimilar to preterm human infants who are surfactant deficient), we delivered Curosurf^®^ (CS; a commercially available surfactant from Chiesi Parma, Italy) IN to mimic the intratracheal (IT) instillation in human babies, followed by AVR-48 injected IP, to demonstrate if AVR-48 has good compatibility with CS when given as adjuvant treatment. There was no change in the total cells or the total protein content in BALF between the BPD groups treated with AVR-48 alone or CS alone or in combination with CS and AVR-48, as compared to untreated BPD group alone (Appendix A).

### 2.6. AVR-48 Increases Lung TLR4 Expression

From in silico molecular modeling and in vitro studies, we found that AVR-48 has a binding affinity for toll-like receptor (TLR) 4 and therefore decreases the expression of TLR4 level in THP-1 human monocytic cells with an EC_50_ of 76.0 nM after 48 h of treatment, as determined by ELISA (unpublished). In the clinical scenario, neonates are vulnerable to infection due to weakened immunity and rely on their innate immune system to combat any externally acquired infection and TLR4 is a crucial component of the neonatal immune system. To determine TLR4 expression in the lung with AVR-48 treatment, Western blot was performed on whole lung homogenates. There was an increase in the expression of TLR4 in the lungs after treatment (Appendix A). Although AVR-48 decreases TLR4 expression in a macrophage cell line (unpublished), it increases TLR4 expression in the lung tissue from our in vivo BPD murine model.

### 2.7. AVR-48 Normalizes Two Important Innate Immune Cell Populations in Animals with BPD

Based on the affinity of AVR-48 for TLR4 and the increased TLR4 expression in AVR-48 treated BPD animals, we questioned if there would be an impact on immune cell recruitment to the lung interstitium. Flow cytometry was performed to determine absolute numbers of key immune cell populations in the lungs of neonatal mice pups from AVR-48-treated and untreated animals in the RA and BPD groups. The cell populations were identified as follows: macrophages (CD45^+^CD11b^+^Ly6G^−^F4/80^+^), dendritic cells (CD45^+^CD11c^+^CD103^+^MHCII^high^), neutrophils (CD45^+^CD11b^+^Ly6G^+^), B cells (CD3^−^CD19^+^), T helper cells (CD3^+^CD4^+^), cytotoxic T lymphocytes (CD3^+^CD8^+^) and NK cells (CD3^−^NK1.1^+^). The gating strategy is provided in Appendix A. These cell populations were chosen to identify both innate and adaptive immune cell populations. First, to determine the impact of AVR-48 alone on immune cell recruitment to the lung, RA animals were compared to AVR-48-treated RA animals. AVR-48-treated RA animals had a slight, but statistically significant decrease in macrophages and increase in dendritic cells in the lung. All other cell populations were similar. (Appendix A). Therefore, AVR-48 by itself had a minimal impact on the immune cell composition in the lung.

In agreement with published literature and prior studies [32,33,34], there was a significant increase in neutrophils [35] and dendritic cells [36] in the BPD group over the RA group (Appendix A). Interestingly, AVR-48 treated BPD animals had decreased neutrophils and increased macrophages compared to untreated BPD animals, and these cell populations were at similar levels as the RA control group. Therefore, AVR-48 normalized two important innate immune cell populations in the setting of BPD.

### 2.8. AVR-48 Suppresses Inflammation in the Lungs by Decreasing the Pro-Inflammatory, and Increasing Anti-Inflammatory Cytokines

Based on this difference in neutrophil and macrophage balance with AVR-48 treatment and the significant improvement in important metrics for BPD severity (Figure 2A), we hypothesized that inflammatory cytokine production would be minimized in the AVR-48 treated animals. Alveolar inflammation is one of the hallmark features in the pathogenesis of BPD. As reported by us [37,38] and others [39], several cytokines and chemokines are upregulated in BPD. Upon treatment with AVR-48, there was a marked decrease in the master inflammatory transcription factor nuclear factor kappa B (NfkB), and pro-inflammatory cytokines tumor necrosis factor (TNF)α, interleukin (IL)-13, and IL-1β, which are otherwise dramatically increased in BPD group, as compared to RA controls in lung homogenates (Figure 7A,B). In contrast, the anti-inflammatory cytokine IL-10 increased upon treatment, which was considerably decreased in BPD group (Figure 7A), as was evident by Western blotting. Similar results were also obtained from lung lysates (data not shown) and blood serum by ELISA assay. Several of the pro-inflammatory cytokines such as monocyte chemoattractant protein (MCP)-1, interferon gamma induced protein (IP)-10, interferon (IFN)γ, IL-1β, and TNFα were significantly upregulated in the serum in the BPD group as compared to RA controls and decreased to normal levels after treatment with AVR-48 (Figure 7C). The RA+AVR-48 group was not included as the samples were not available at the time of conducting these assays. IL-10 was markedly increased after drug treatment in the BPD group (Figure 7C). Together, these data indicate that AVR-48 does not impact immune cell recruitment to the lung in the setting of BPD. However, AVR-48 effectively modulates inflammatory and anti-inflammatory cytokine production, shifting treated animals in favor of an anti-inflammatory environment, which potentially improves the lung morphometry.

### 2.9. AVR-48 Protects the Lungs from Progressing toward BPD-PH

BPD-PH is characterized by abnormal vascular remodeling and rarefication of the pulmonary vasculature leading to vascular growth arrest, which eventually leads to increased pulmonary vascular resistance and right heart failure [40]. A similar effect is also seen in mouse models of experimental BPD, as reported by us previously [24,33,34,41]. There is hypertrophy of the right ventricle (RV) with an increase in the thickness of interventricular septum (IVS). RV and left ventricle (LV) and Fulton’s Index (RV/LV+IVS) is higher in the BPD group as compared to RA, but is decreased significantly after treatment with AVR-48 in the BPD group (Figure 8A). There was no change in the ratio in the RA group treated with the drug, which was similar to RA control group (Figure 8A). Total vascular endothelial growth factor (Vegf) was less in the BPD group, which increased considerably after treatment with AVR-48 (Figure 8B); conversely, endothelial nitric oxide synthase (eNOS) [42] and bone morphometric protein receptor (BmpR)II, which are known to be elevated in BPD [43,44], were significantly decreased in the AVR-48 treated group (Figure 8C). Vegf-D has not been reported earlier in mouse models of BPD, to the best of our knowledge, and in the present study we report an increase in the expression of this protein in BPD, which decreases after treatment. All the above data show that AVR-48 is able to rescue the BPD cardiopulmonary phenotype.

## 3. Discussion

Despite several advances in neonatal lifesaving methodologies, BPD continues to be one of the most devastating life-threatening conditions in preterm babies. Repeated inflammatory insults from antenatal complications and postnatal consequences worsen the lung phenotype. BPD-PH further contributes significantly to severe morbidity and mortality. However, if prevented early, this condition can be mitigated to improve the respiratory status and developmental delays in childhood of these preterm infants. We have recently reported that AVR-48 decreases severe lung inflammation in LPS-, hyperoxia- and CLP-induced ARDS in adult mice while AVR-25, another close analog of AVR-48, can prevent lung injury in the neonatal mouse model of experimental BPD [19]. In order to facilitate bulk manufacturing, and improve physicochemical properties, we have optimized the chemical series and identified AVR-48 as a better lead than our previously reported compound, AVR-25. In the present study, we report that, AVR-48 is able to prevent experimental BPD by alleviating lung injury and BPD-PH. As BPD is a neonatal disease, we made every effort to make our compound nontoxic by synthesizing it in the purest form during formulation and noted that a high dose of 100 mg/kg did not have any toxic effects in the visceral organs. Most importantly, from the dose response efficacy and toxicokinetic studies, we determined that the therapeutic index for AVR-48 in the juvenile mouse was 20-fold, based on the minimum efficacious dose of 5 mg/kg/day vs. NOAEL of 100 mg/kg/day, which is a highly desirable profile for a lead drug candidate.

As surfactants are the standard of care in NICUs all over the world, we tested AVR-48 administered adjuvantly with CS in mouse pups to rule out any cross reactivity with surfactant, in vivo. There was no adverse effect on the interaction between the drug and CS. The chord length was normal and total cells and protein content in the BALF were similar to that of the groups treated with AVR-48 alone or with CS alone. Additional state-of-the-art morphometry using stereology [45] may provide improved delineation of the characteristic pulmonary phenotype of experimental BPD.

Increased inflammation, decreased or dysregulated angiogenesis, increased cell death and decreased cell proliferation are certain key features associated with the pathogenesis of BPD. Total alveolar cell proliferation was decreased in the BPD group, as was evident from the number of Ki67+ve (Figure 4A) and PCNA+ve (Figure 4B,C) cells. There was a marked decrease in the number of SP-C^+^PCNA^+^ cells in the BPD group as compared to RA, RA+AVR-48, and BPD+AVR-48 groups. Although the number of RAGE+ve cells were less in the BPD than RA and RA+AVR-48 groups, there were few cells which were double positive for PCNA as well as RAGE in all four groups. Thus, our results suggest that Type II AECs (SP-C+ve) cells were more proliferative than Type I AECs (RAGE+ve) cells, irrespective of hyperoxia treatment. Work from several investigators in the past have reported that differentiation of AECs from Type II to Type I is associated with decreased epithelial cell proliferation [46]. Type II AECs proliferate following hyperoxia exposure in several animals including mice and baboons [46], but later decrease in number during the recovery phase [47]. It is known that hyperoxia exposure kills the majority of Type I AECs in adults [48]. In the present study, upon treatment with AVR-48, the proliferation of SP-C+ve Type II AECs were increased, while there were fewer cells double positive for PCNA and RAGE.

AVR-48 was able to suppress inflammation by inhibiting TGFβ, NFkB, TNFα, IL1β, MCP-1, IP-10, and IFNγ—all of which are mediators of inflammation; in contrast, the anti-inflammatory cytokine IL-10 was upregulated upon treatment. Although AVR-48 has a binding affinity for TLR4, there was increased TLR4 expression after AVR-48 treatment. TLR4 is activated following hyperoxia exposure in the neonatal brain [49] and lung [50], which leads to inflammatory cytokine release. Potential binding of AVR-48 to TLR4 may activate the TIR-domain-containing adapter-inducing the interferon-β (TRIF) pathway, which results in increased production of IL-10 and decreased production of MyD88-dependent inflammatory cytokines, such as TNFα and IL1β [51]. IL-10 acts as a suppressor of TLR4 and thus this increased IL-10 may serve as a negative feedback loop for TLR4 activation [52]. From the results of our present study, we speculate that AVR-48 may act as a feedback modulator as a result of which the binding of AVR-48 with TLR4 enhances resident or anti-inflammatory macrophages M2 over inflammatory macrophages M1 via an alternate pathway activation, as has been shown with chitohexaose in an LPS-induced sepsis model [21]. Therefore, we propose that AVR-48 may act as a feedback modulator that acts through the TRIF pathway and initiates production of IL-10. This IL-10 production then suppresses TLR4 and downregulates MyD88-dependent production of pro-inflammatory cytokines (Figure 9). Further mechanistic studies are needed to prove this hypothesis.

PH is often associated with BPD and this condition has also been observed in mice models of experimental BPD. Vegf, which is considered a classical marker for BPD-PH [53], was decreased while eNOS was increased in BPD. For the first time, in this study we report that Vegf-D, which is a lymphangiogenic growth factor, is increased in BPD. Both eNOS and Vegf-D are substantially decreased after treatment with AVR-48. The significance of this decrease needs further study to emphasize the role of Vegf-D in BPD-PH.

Mutations in BmpRII are associated with heritable pulmonary arterial hypertension (PAH) and there are multiple reports that show that in adult PAH, there is a decrease in BmpRII. However, the role of BmpRII in neonatal hyperoxia and BPD-PH has received less attention. While Alejandre-Alcázar et al. showed that *BmpRII* is significantly decreased after hyperoxia exposure (85% O_2_ for 14 or 21 days) in P10 or P14 mouse pups [43], Chen et al. showed that *BmpRII* is significantly increased in rat pups (P10) after hyperoxia (100% O_2_) for 10 days exposure [44]. Yee et al. reported that *BmpRII* decreases in the adult mice when these mice were exposed to hyperoxia in the neonatal stage; they have no data regarding the expression of *BmpRII* after hyperoxia exposure in the neonatal stage [54]. All three groups have reported only the mRNA expression of *BmpRII* because the antibodies for BmpRII did not work to evaluate the same protein levels. In contrast, we have shown that the protein expression of BmpRII is increased following hyperoxia exposure, which was subsequently downregulated after treatment with AVR-48. In many instances, the mRNA levels do not correlate with the protein levels. Hence, the role of BmpRII in hyperoxia-induced BPD-PH needs to be evaluated more elaborately in the neonatal context.

Based on the results of this study, we can suggest that AVR-48 has the potential to be developed commercially as a prophylactic therapy for an orphan disease such as BPD and BPD-PH, for which there is no prevention or cure to date.

## 4. Materials and Methods

### 4.1. Animals

C57BL/6 J mice were purchased from The Jackson Laboratory (Bar Harbor, ME, USA) and were maintained in a breeding colony at Drexel University, Philadelphia, PA, USA. Animal procedures were performed in accordance with the NIH *Guide for the Care and Use of Laboratory Animals* and were approved by the Institutional Animal Care and Use Committee (IACUC) of Drexel University Philadelphia, PA (Protocol No. 20706, approved on July 7, 2015). Neonatal rat pups born from female pregnant Sprague Dawley Crl:CD (SD) rats (Charles River Laboratories, St-Constant, QC, Canada) were used for the toxicology and PK studies to evaluate safety, determine the maximum tolerated dose (MTD), and the PK profile of AVR-48. A minimum 6-day acclimation period was allowed between receipt of the animals and the start of treatment to accustom the rats to the laboratory environment. All rat studies were approved by the IACUC of ITR Laboratories, Montreal, QC, Canada (Protocol No. 74691, approved on Aug 10, 2019). The sex, number of rat pups, and experimental conditions for each experiment (PK studies) are summarized in Appendix A. Attempts were made to use the whole litter born (*n* = 6) for mouse pups for any given experimental assay. Mice pups were used for morphology, pharmacology, toxicology, biochemistry, and molecular biology studies while rat pups were used to confirm morphology, pharmacokinetics, and toxicology studies.

### 4.2. Chemicals and Reagents

The synthesis and structural characterization of compound AVR-48 and the PLGA encapsulated AVR-48 were conducted in the laboratory of AyuVis Research Inc. (Fort Worth, TX, USA), following their in-house procedures [55]. The synthesis, characterization, and drug release of AVR-48 nanoparticle suspension is provided in the Appendix A. Endotoxin-free phosphate-buffered saline (PBS) was purchased from Sigma-Aldrich Inc., St. Louis, MO, USA.

### 4.3. Formulation of AVR-48 for Efficacy and Toxicokinetic Studies

For the mouse dose response efficacy studies, AVR-48 was reconstituted in 0.9% sterile normal saline to provide a final dose concentration of 1.0 mg/kg, 2.5 mg/kg, 5.0 mg/kg, and 10 mg/kg as a colorless solution, and injected IP (30 µL) on P2 and P4. The PLGA encapsulated AVR-48 or the GFP-tagged analog was resuspended in deionized sterile water to create a nanosuspension with final dose concentration of 0.025, 0.05, and 0.11 mg/kg/drop. The surfactant Curosurf^®^ (Cheisi Parma, Italy), available commercially, was delivered IN at a volume of 3µL per nostril, on P2 and P4. For the toxicokinetic study of AVR-48 in rat pups, a formulation of 10% DMSO, 20% tetraglycol, 20% PEG 400, and 50% sterile water was made fresh before administration [22].

### 4.4. Hyperoxia Treatment

For the hyperoxia experiment, newborn pups were exposed to 100% hyperoxia (P0), along with their mothers, in cages in an airtight Plexiglas chamber (OxyCycler; Biospherix, Redfield, NY, USA) as described previously by us [19,24,34] for 4 consecutive days (till P4) and removed to RA on P5 to be recovered till P14, so as to emulate the human BPD condition. All pups were sacrificed on P14 for further experimental analyses. The pups without any hyperoxia exposure served as the corresponding RA (normoxia) controls. The lactating mothers were alternated every 24 h with pups in room air conditions to ensure proper nutrition (milk supply) to the pups and to avoid oxygen toxicity to the mothers.

### 4.5. Bronchoalveolar Lavage Fluid (BALF) Analysis

Pups were sacrificed on PN14, and the trachea was cannulated with a small-caliber needle by instilling PBS endotracheally at 25 cm H_2_O pressure for 15 min. Two volumes of 300 μL of cold 1× PBS were instilled, gently aspirated, and pooled. Samples were centrifuged at 1000× *g* for 10 min at 4 °C. The supernatant was collected, and total protein was quantified using the Pierce^TM^ BCA Protein Assay Kit (Fisher Scientific Co, Houston, TX, USA). The total cell count was performed using the TC20 cell counter (BioRad, Hercules, CA, USA).

### 4.6. Histology, Immunohistochemistry, and Immunofluorescence

Both the RA control and BPD mice were anesthetized (using an overdose of a cocktail of xylazine-ketamine) and lung and heart tissues were harvested after perfusion and fixed overnight in 4% paraformaldehyde. Fixed tissues were then washed in fresh PBS, and stored in 70% ethanol before being processed in an autostainer (Ventana, CA, USA) by the Histology Core Facility of Wistar Institute, Philadelphia to be stained with hematoxylin and eosin (H&E) for lung morphometry or immunohistochemistry/immunofluorescence as previously described [19,23,24]. Immunofluorescence staining was performed for Ki67 (Abcam, MA, USA 1:10) SP-C (Abcam, MA, USA, 1:100), RAGE (R&D Systems, Minneapolis, MN, USA, 1:100), and Von Wilebrand Factor (vWF-DAKO, 1:100) on lung paraffin sections following the protocol as described earlier [24], while TUNEL staining was performed following the manufacturer’s instructions (Roche Diagnostics, Indianapolis, IN, USA).

### 4.7. Morphometry and Quantification

To study chord length, septal thickness, and radial alveolar count in the lungs, RV, LV, and IVS thickness, 5 µm thick left-lobe lung and heart paraffin-embedded sections were stained with H&E. Multiple randomly chosen areas (at least 10 areas) from each section were photographed using 100× total magnification. Chord length and septal thickness were measured by installing the respective plugins (Shift+C) and (Shift+S; the plugin for bone trabecular thickness) in the ImageJ software [56]. The lung morphometry parameters were measured as described [19,23,24,25] either using ImageJ (a free software of NIH) [56] or CellSens software [57]. Images of H&E stained slides were taken at total magnification of ×100 for the software to calculate the mean distance between the airspaces (for chord length) and the free ended septal thickness in every field (for septal thickness) using the plugins as mentioned above. Sections with large airways or blood vessels from the lung were excluded while imaging the slides for lung morphometry. Radial alveolar count (RAC), which measures the complexity of the terminal respiratory unit (acinus), was assessed following the methodology of Emery and Mithal [58], in which the number of alveoli are counted from a respiratory bronchiole to the edge of the acinus. Using CellSens software [57], a perpendicular line was dropped from the center of the bronchiole to the edge of the acinus (pleura), and the number of alveoli cut by this line was then counted manually.

Quantitative measurements of PH–induced RV hypertrophy ratios (RV/LV and RV/LV+IVS) were performed using the methodology described previously [41]. CellSens software [57] was used to measure the right ventricular wall thickness by drawing an arbitrary line, and recording the thickness of the wall, which is displayed automatically by the software.

For quantification of cell proliferation and cell death, the entire lung section was divided into three areas, and the total number of Ki67+ve and TUNEL+ve cells nuclei were counted manually, which was normalized with the total number of nuclei to give a percentage of positive cells. For vWF quantification, total number of closed vessels were counted per high power field area in one lung section. A minimum of three areas was chosen, and three to seven animals were used for staining and counting.

### 4.8. Western Blot Analysis

Western blot analyses for TLR4, total Caspase 3, and cleaved Caspase 3 (Cell Signaling Technology, Danvers, MA, USA; 1:1000), Ang2 (1:500; Sigma, St. Louis, MO, USA), TGFβ, NFkB, TNFα, IL-10, IL-1β, IL-4, Vegf, eNos, Vegf-D, BmpRII, and Vinculin (1:500; SantaCruz, Dallas, TX, USA) were performed, as previously described [24], by loading 30 µg of lung protein, followed by immunoblotting with the above antibodies and visualizing with Odyssey infrared imaging system (LI-COR Biosciences, Lincoln, NE, USA). Densitometric quantification was performed using ImageJ after normalizing with Vinculin, the loading control housekeeping protein.

### 4.9. Multiplex ELISA

Lung lysate and blood serum from control and AVR-48 treated group were used for Multiplex ELISA and performed on four separate inflammatory panels of Meso Scale Discovery multispot assay system (MSD, Rockville, MD) to detect two chemokines (MIP-2, MCP-1) and eight cytokines: IL-21, IP-10, IFNγ, IL-1β, TNFα, IL-17, IL-10, and IL-6, following the manufacturer’s instructions. Briefly, serum samples and lung were diluted 1:1 in a total volume of 25µL (for a final concentration of 10 µg of lysate/well) with the dilution buffer provided with MSD kit and incubated with the above labeling antibodies for 2 h, RT followed by washing with PBST. The absorbance was detected using the MSD-specific luminometer.

### 4.10. Imaging

All images were captured on an Olympus IX70 with DP73 camera attachment. At least five to seven images (magnifications of ×10 or ×20 or ×40, as and when appropriate) were acquired for quantification. CellSens software [57] was used for capturing of images and further modified with Adobe Photoshop [59] for acquiring the best images.

### 4.11. Statistical Analysis

All statistical analyses were performed using GraphPad Prism [60]. The data are expressed as the mean ± SEM with *n* = 5 to 7 mice in each group. Groups were compared with the two-tailed unpaired t-test and one- or two-way analysis of variance (ANOVA), as appropriate. *p* ≤ 0.05 was considered statistically significant. For the toxicology and PK study, a generalized analysis of variance/covariance (ANOVA/ANCOVA) test was performed on the numerical data (three or more animals/groups) on the study as follows: An automatic transformation was used to analyze the data for homogeneity of variance using Levene’s test. Parametric and non-parametric trends were analyzed using the Williams and the Shirley-Williams tests, respectively. Homogeneous data was analyzed using the ANOVA/ANCOVA, and the significance of intergroup differences between the control and test item-treated groups was analyzed using Dunnett’s test. Heterogeneous data were analyzed using Kruskal-Wallis test and the significance of intergroup differences between the control and test item-treated groups was assessed using a nonparametric Dunnett’s test. All data are reported as ± SEM. A significance level of *p* < 0.05 at 95% confidence intervals was considered statistically significant for all the experiments reported in this study.

## Figures and Tables

**Figure 1 ijms-22-08547-f001:**
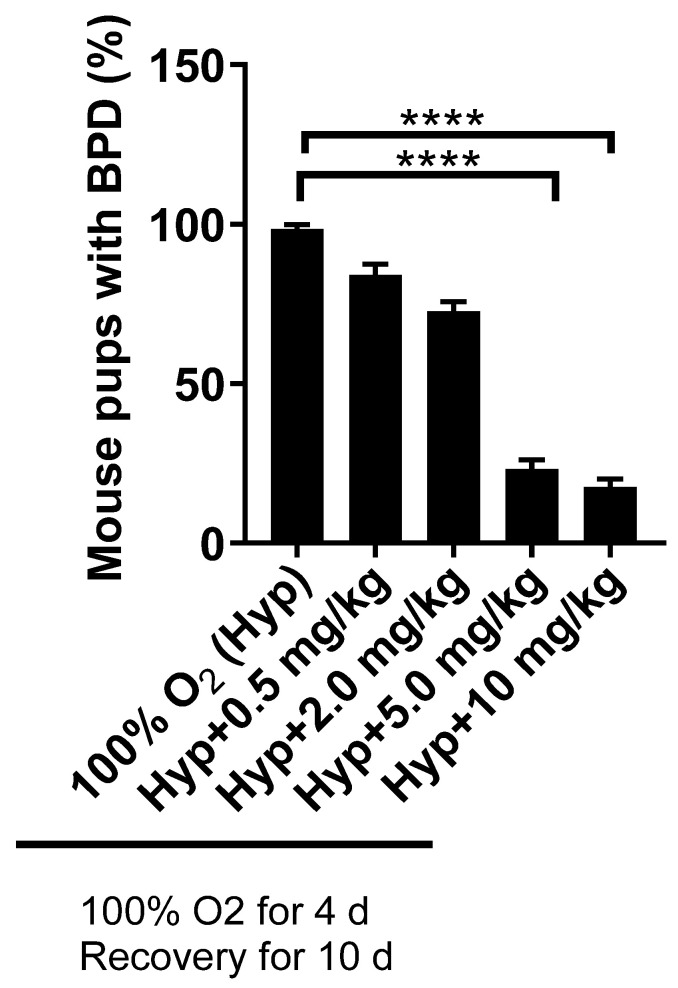
Dose response study of AVR-48 at different doses of 0.5, 2.0, 5.0, and 10 mg/kg, given IP. **** *p* < 0.0001. *n* = 5–7 mice per group. 10 mg/kg (IP) was selected as the most efficacious dose. IP: intraperitoneal; Hyp: hyperoxia; BPD: bronchopulmonary dysplasia.

**Figure 2 ijms-22-08547-f002:**
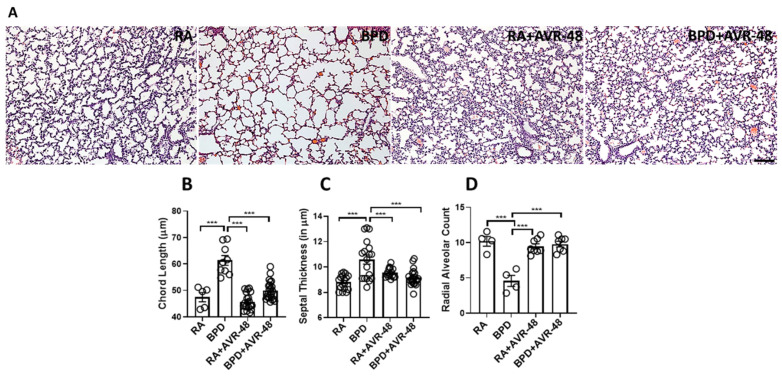
AVR-48 improves lung morphology. (**A**) Representative H&E stained lung paraffin sections showing histological changes after AVR-48 treatment. (**B**) Chord length (which measures the average free distance in the air spaces) is increased in the BPD group and normalizes after AVR-48 treatment. (**C**) The alveolar septal thickness is decreased and (**D**) the radial alveolar count (which measures the number of alveoli) is also improved after AVR-48 treatment. *******
*p* < 0.001, *n* = 3–8; RA: room air; BPD: bronchopulmonary dysplasia. Scale bar 100 µm.

**Figure 3 ijms-22-08547-f003:**
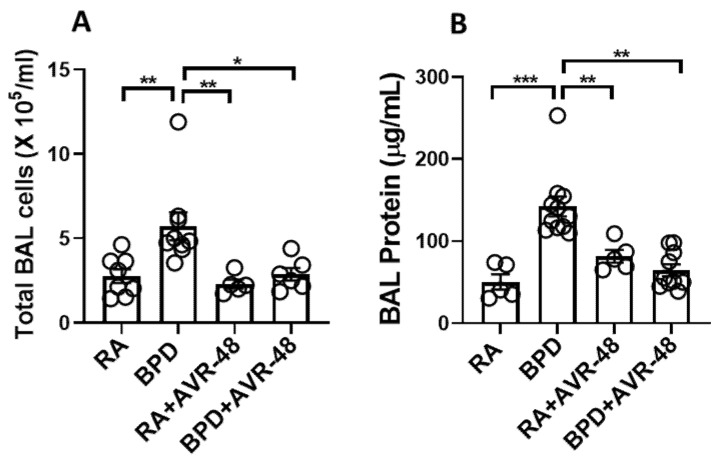
AVR-48 decreases inflammation and vascular leak. (**A**) Total inflammatory cells in the BAL fluid in the BPD group is significantly decreased after AVR-48 treatment. (**B**) Total protein in the BAL fluid in the BPD group is significantly decreased after AVR-48 treatment. * *p* < 0.05; ** *p* < 0.01; *** *p* < 0.001, *n* = 3–8; RA: room air; BAL: bronchoalveolar lavage; BPD: bronchopulmonary dysplasia.

**Figure 4 ijms-22-08547-f004:**
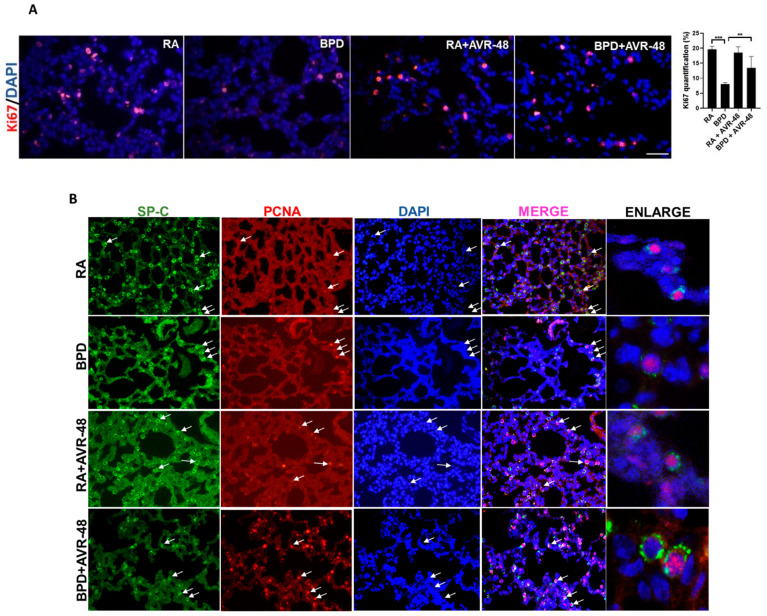
AVR-48 improves cell proliferation (**A**) AVR-48 treatment in the BPD group increases cell proliferation (as shown by Ki67 staining) and the right panel shows quantification for Ki67. (**B**) Co-localization of SP-C (marker for Type II AECs) with PCNA. White arrows point to the respective cells that are proliferating. Extreme right panel shows higher magnification of proliferating Type II AECs positive for SP-C (cytoplasmic green) and PCNA (nuclear red). (**C**) Co-localization of RAGE (marker for Type I AECs) with PCNA. Extreme right panel shows higher magnification of proliferating Type I cells positive for RAGE (cytoplasmic green) and PCNA (nuclear red). ** *p* < 0.01; *** *p* < 0.001; Scale bar 100 µm. RA: room air; BPD: bronchopulmonary dysplasia; SP: surfactant protein; AECs: alveolar epithelial cells; PCNA: proliferating cell nuclear antigen; RAGE: receptor for advanced glycation end products.

**Figure 5 ijms-22-08547-f005:**
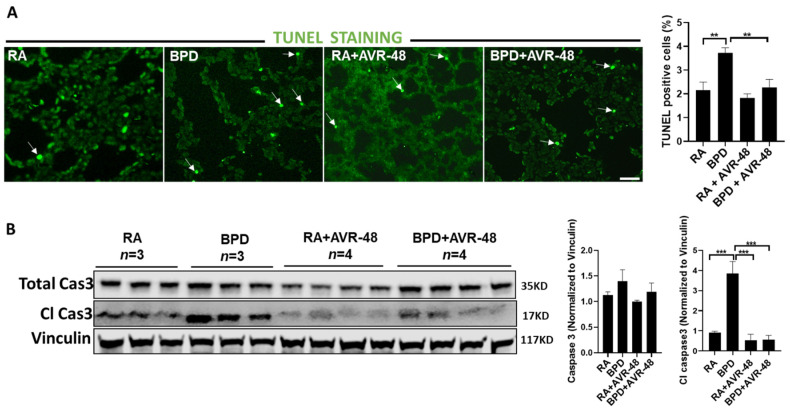
AVR-48 decreases cell death: TUNEL staining (white arrows point to TUNEL positive cells) (**A**) and Western blotting of total caspase 3 and cleaved caspase 3 (**B**) shows decrease in cell death and apoptosis after treatment with AVR-48. Right panel shows quantification of TUNEL positive cells (top) and densitometric quantification of total caspase 3 and cleaved caspase 3 (bottom). *n* = 3–4. ** *p* < 0.01; ****p* < 0.001; Scale bar 100 µm. RA: room air; BPD: bronchopulmonary dysplasia; Cl Cas: cleaved caspase.

**Figure 6 ijms-22-08547-f006:**
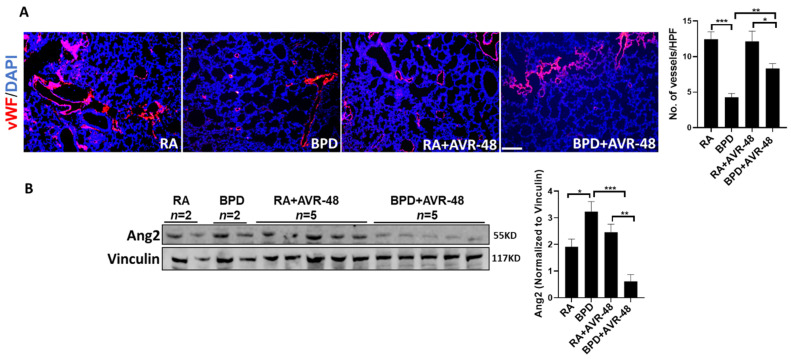
AVR-48 promotes vascular development. (**A**) Representative immunofluorescent lung sections showing vascular development. vWF, a marker for blood vessels, is severely disrupted in BPD, while after treatment with AVR-48 there is significant improvement. (**B**) Representative Western blotting showing Ang2 is restored after treatment with AVR-48, in the BPD group. The top right panel shows quantification of the number of blood vessels while the bottom right panel shows densitometric quantification for Ang2. Scale bar 100 µm; * *p* < 0.05; ** *p* < 0.01; *** *p* < 0.001; vWF: von Willebrand factor; Ang2: angiopoietin 2; RA: room air; BPD: bronchopulmonary dysplasia; HPF: high power field; *n* = 3–5.

**Figure 7 ijms-22-08547-f007:**
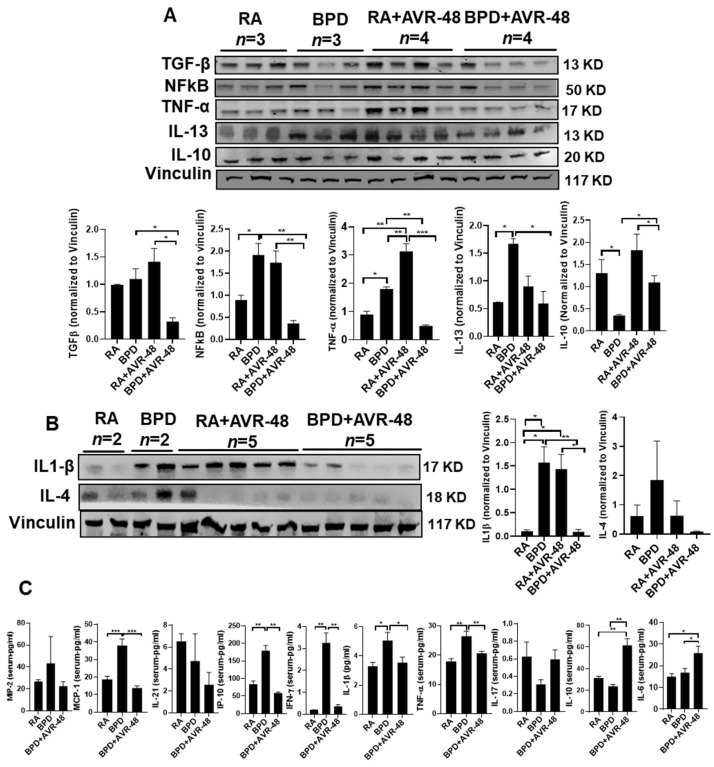
AVR-48 suppresses inflammation. (**A**,**B**) Representative Western blot showing decrease of pro-inflammatory cytokines (TGFβ, NFkB, TNFα, IL-13, IL-1β, and IL-4) and increase of IL-10 in the lungs after treatment with AVR-48, as compared to the BPD group. The increased inflammation seen in the RA+AVR-48 treated group may be due to the natural defense adaptive mechanism. Vinculin is the loading control. The panel below the gel shows densitometric quantification of the proteins. *n* = 5. (**C**) ELISA showing the expression of selected cytokines in the blood serum of treated BPD group as compared to untreated BPD controls. The RA+AVR-48 group was not included for this assay. Although most of the pro-inflammatory cytokines and chemokines show a decrease after treatment, there was no change in MIP-2, IL-21, or IL-17. * *p* < 0.05; ** *p* < 0.01; *** *p* < 0.001, *n* = 4–5; RA: room air; BPD: bronchopulmonary dysplasia; TGFβ: transforming growth factor beta; MCP-1: monocyte chemoattractant protein 1; MIP-2: macrophage inflammatory protein 2; NfkB: nuclear factor kappa B; TNFα: tumor necrosis factor alpha; IFNγ: interferon gamma; IL: interleukin.

**Figure 8 ijms-22-08547-f008:**
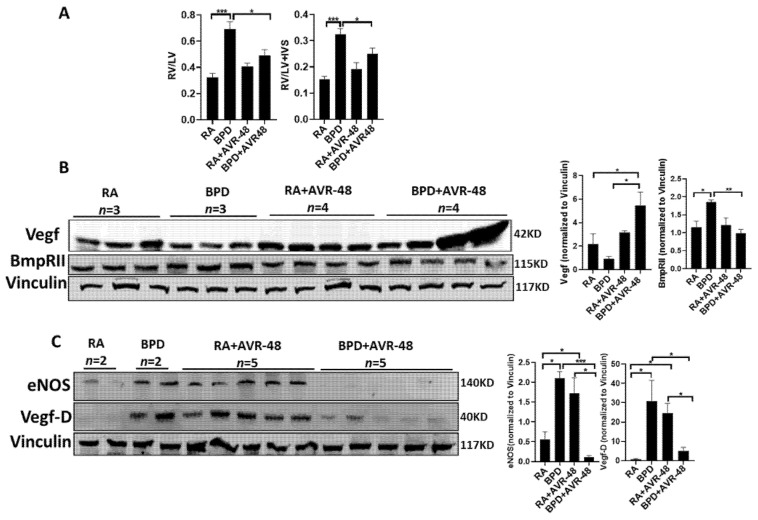
AVR-48 protects against BPD-PH. (**A**) The RV/LV ratio and Fulton’s Index (RV/LV+IVS) is improved after AVR-48 treatment in the BPD group. (**B**) Representative Western blot showing an increased expression of Vegf in the BPD+AVR-48 treated group as compared to BPD group. (**C**) eNOS, BmpRII and VegfD, which are increased in BPD, are noticeably decreased after treatment with AVR-48. Vinculin is the loading control. As the same samples were used for Figure 7A,B and Figure 8B,C, the same vinculin has been shown for both images as the loading control. *n* = 4–5. BPD-PH: bronchopulmonary dysplasia-associated pulmonary hypertension; RV: right ventricle; LV: left ventricle; IVS: interventricular septum; RA: room air; BPD: bronchopulmonary dysplasia; Vegf: vascular endothelial growth factor; eNOS: endothelial nitric oxide synthase; BmpRII: bone morphogenetic protein receptor 2. * *p* < 0.05; ** *p* < 0.01; *** *p* < 0.001.

**Figure 9 ijms-22-08547-f009:**
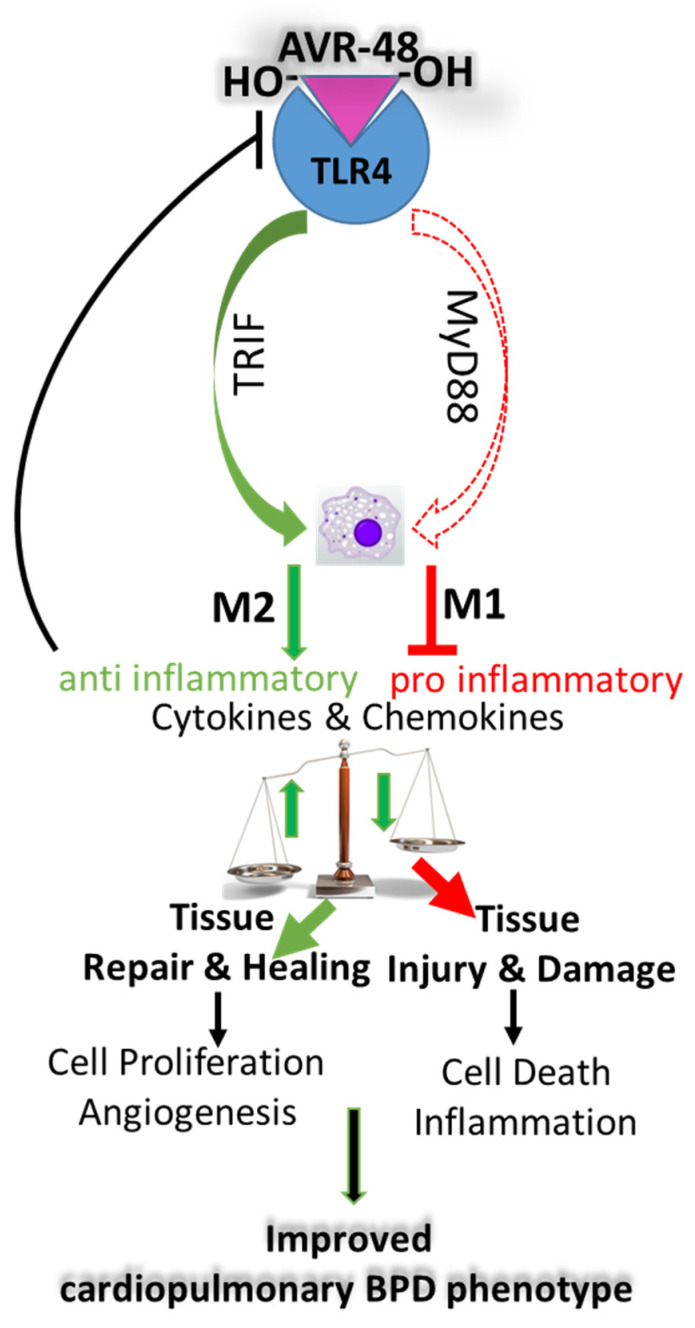
Proposed mechanism of action of AVR-48 in neonatal lungs. AVR-48 after binding to TLR4 triggers the TRIF pathway to activate the M2 macrophages via the alternate pathway to produce IL-10, which in turn negatively regulates TLR4 to downregulate the MyD88 pathway to decrease the synthesis of a myriad of pro-inflammatory cytokines and chemokines by suppressing the M1 macrophages that are produced via activation of the classical pathway during BPD. This combinatorial effect results in decreasing tissue injury and increasing tissue repair and healing by maintaining a balance between M2 and M1 macrophages toward a favorable outcome with overall improvement of the BPD cardiopulmonary phenotype.

## Data Availability

Raw uncut gels of the Western Blots in the “results” are shown in Appendix A.

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
