# Peer review of "Chitin-Derived AVR-48 Prevents Experimental Bronchopulmonary Dysplasia (BPD) and BPD-Associated Pulmonary Hypertension in Newborn Mice"

_ijms, 2021, doi:10.3390/ijms22168547_

Round 1
Reviewer 1 Report
The manuscript “Chitin-derived AVR-48 prevents Experimental Bronchopulmonary Dysplasia (BPD) and BPD-associated Pulmonary Hypertension in Newborn Mice” by Bhandari et al. shows an elaborate set of in vivo and in vitro experiments characterizing the pharmacokinetic and toxicologic profile of the chitin-derived AVR-48 and its anti-inflammatory and pro-angiogenic effect in preventing BPD and PBD-associated PH. They used bronchoalveolar lavage fluid, lung homogenate western blot and ELISA, lung immunohistology samples, lung cell flow cytometry, and serum ELISA to show the pro-angiogenic and anti-inflammatory markers improved AVR-48 in newborn mice. If they had used primary human cell (neonatal pulmonary cell) lines to show angiogenic and proliferative effects (like tubule formation assay, MTT, or BrdU assays), it would be more contextual as a future drug candidate for premature babies. Yet, I support the publication of this paper with some minor suggestions.
- Page 2, line 64- “There have been no randomized clinical trials (RCTs) where inhaled budesonide has been used to treat ‘established BPD’ [6]” was immediately followed by “In the largest RCT on inhaled budesonide [7]……….. These two sentences look contradicting to each other. Budesonide is again discussed at the end of the following paragraph, line 77. It would be easier for the reader to follow if BPD treatment approaches are put together in a separate paragraph rather than at the end of the first three paragraphs of the introduction.
- Page 8, line 283 – “Although AVR-48 decreases TLR4 expression in a cell line…... needs a reference.
- Page 14, line 503 “….. dehydrated using 70% ethanol, cleared….” Usually, the dehydration process starts with 70% ethanol with successively increasing the percentage to 100% ethanol (to remove all moisture content) before clearing with xylene (insoluble to water).
- Page 15, line 538- “Briefly, samples were diluted 1:1 in a total volume of 25 µL….” Please state the total amount of protein (how many micrograms of protein) used per sample or per well.
- Western blot images should state the molecular weight of the corresponding proteins.
Author Response
Please find attached the point by point address of the comments.

Reviewer 2 Report
The manuscript reports extensive experiments that test the hypothesis that an anti-inflammatory chitin derivative (AVR-48) alleviates pathological alterations associated with an experimental BPD mouse and rat model. Although at first glance the manuscript seems to report a plethora of thorough analyses and sound data there are some hints that raise doubts about the overall quality of the work:
- Figure 1: How do the authors define a mouse pup with BPD? This seems to me not an objective measure, particularly given the small number of n=5-7 per group.
- The authors do not mention how the number of animals per group was chosen for any of the experiments? How did the authors adjust for differences between different litters? Did the mother animal stay in hyperoxia during the whole exposure? This might cause secondary effects. n=3-8 in some figures is really not a suitable information. At best, the authors would show dot blots to provide the data of each animal.
- The capillaries of alveolar septa do not contain von Willebrand factor. However, the authors show IHC of this in Fig. 5. The "rescued" vWF staining in fig 5 (BPD+AVR-48) is an artefact, not a staining result.
- From this, I would like to know how the authors tested the specificity of the many antibodies used.
- The quality of the microscopic images is very poor and hardly anything can be seen here.
- The morphometric methods are not described in detail. Although they are themselves not state of the art (neither the chosen parameters nor the preparation of the tissue and the sampling design), it would be mandatory to give the reader a chance of knowing what the authors have done.
- Sometimes it is not clearly said what was done in the mouse and what in the rat. Why were both species actually used? Although their lung biology is similar it is not the same. I would also like to know how the authors managed to inject the AVR-48 intravenously in a mouse pup at P2 and how sure they are that they injected IV. Would love to see some evidence for this.
- Ki-67: Without co-stainings to know which cells are proliferating this has no meaning.
- TUNEL: This is not an acceptable method of apoptosis as a single measure of apoptosis.
- Imprecise language at some points, e.g. page 5, l. 232: "blood vessels, which are usually disrupted in BPD" What does that mean?
- The authors talk about their animals as if they had BPD. This is only a model of BPD. And it is not a very commonly used one. 100% hyperoxia is probably much different from human BPD.
Author Response

(The authors gave the same response as above.)

Round 2
Reviewer 2 Report
The authors have done a great job in responding to my concerns and revising the manuscript which I didn't expect. For future work, however, I would like to recommend to use more state-of-the-art morphometry, i.e. stereology. There is a large body of literature on this and I think the authors might be convinced that the methods used in the manuscript to describe the stunted alveolarization in BPD are not useful at all although they are widely used. I think this issue should be mentioned in the discussion.
Author Response
Response to Reviewer’s comments:
Reviewer 2
C1: The authors have done a great job in responding to my concerns and revising the manuscript which I didn't expect. For future work, however, I would like to recommend to use more state-of-the-art morphometry, i.e. stereology. There is a large body of literature on this and I think the authors might be convinced that the methods used in the manuscript to describe the stunted alveolarization in BPD are not useful at all although they are widely used. I think this issue should be mentioned in the discussion.
R1: Thank you for the positive comments on the additional work done in our revised manuscript. We agree that additional state-of-the-art morphometry using stereology could provide improved delineation of the characteristic pulmonary phenotype of experimental BPD. This has now been mentioned in the “discussion” of the revised manuscript.